# Capturing an initial intermediate during the P450nor enzymatic reaction using time-resolved XFEL crystallography and caged-substrate

Takehiko Tosha[1], Takashi Nomura[1], Takuma Nishida[2], Naoya Saeki[2], Kouta Okubayashi[2], Raika Yamagiwa[1,2], Michihiro Sugahara[1], Takanori Nakane [3], Keitaro Yamashita [1], Kunio Hirata[1,4], Go Ueno[1], Tetsunari Kimura[5], Tamao Hisano[1], Kazumasa Muramoto[2], Hitomi Sawai[2], Hanae Takeda[1,2], Eiichi Mizohata[6], Ayumi Yamashita[1], Yusuke Kanematsu[7], Yu Takano[7], Eriko Nango[1,8], Rie Tanaka[1], Osamu Nureki[3], Osami Shoji[9,10], Yuka Ikemoto[11], Hironori Murakami[11], Shigeki Owada[1], Kensuke Tono[11], Makina Yabashi [1], Masaki Yamamoto[1], Hideo Ago[1], So Iwata[1,8], Hiroshi Sugimoto[1,10], Yoshitsugu Shiro[1,2] & Minoru Kubo [1,4]

Time-resolved serial femtosecond crystallography using an X-ray free electron laser (XFEL) in conjunction with a photosensitive caged-compound offers a crystallographic method to track enzymatic reactions. Here we demonstrate the application of this method using fungal NO reductase, a heme-containing enzyme, at room temperature. Twenty milliseconds after caged-NO photolysis, we identify a NO-bound form of the enzyme, which is an initial intermediate with a slightly bent Fe-N-O coordination geometry at a resolution of 2.1 Å. The NO geometry is compatible with those analyzed by XFEL-based cryo-crystallography and QM/MM calculations, indicating that we obtain an intact $Fe^{3+}$-NO coordination structure that is free of X-ray radiation damage. The slightly bent NO geometry is appropriate to prevent immediate NO dissociation and thus accept $H^-$ from NADH. The combination of using XFEL and a caged-compound is a powerful tool for determining functional enzyme structures during catalytic reactions at the atomic level.

[1] RIKEN SPring-8 Center, 1-1-1 Kouto, Sayo, Hyogo 679-5148, Japan. [2] Department of Life Science, Graduate School of Life Science, University of Hyogo, 3-2-1 Kouto, Kamighori, Akoh, Hyogo 678-1297, Japan. [3] Department of Biological Sciences, Graduate School of Science, The University of Tokyo, 2-11-16 Yayoi, Bunkyo-ku, Tokyo 113-0032, Japan. [4] Japan Science and Technology Agency, PRESTO, 4-1-8 Honcho, Kawaguchi, Saitama 332-0012, Japan. [5] Department of Chemistry Graduate School of Science, Kobe University, 1-1 Rokkodai, Nada-ku, Kobe 657-8501, Japan. [6] Department of Applied Chemistry Graduate School of Engineering, Osaka University, 2-1 Yamadaoka, Suita, Osaka 565-0871, Japan. [7] Graduate School of Information Sciences, Hiroshima City University, 3-4-1 Asa-Minami-ku, Hiroshima 731-3194, Japan. [8] Department of Cell Biology Graduate School of Medicine, Kyoto University, Yoshidakonoe-cho, Sakyo-ku, Kyoto 606-8501, Japan. [9] Department of Chemistry Graduate School of Science, Nagoya University, Furo-cho, Chikusa-ku, Nagoya 464-8602, Japan. [10] Japan Science and Technology Agency, CREST, 5 Sanbancho, Chiyoda-ku, Tokyo 102-0075, Japan. [11] Japan Synchrotron Radiation Research Institute, 1-1-1 Kouto, Sayo, Hyogo 679-5198, Japan. Takehiko Tosha, Takashi Nomura and Takuma Nishida contributed equally to this work. Correspondence and requests for materials should be addressed to H.S. (email: sugimoto@spring8.or.jp) or to Y.S. (email: yshiro@sci.u-hyogo.ac.jp) or to M.K. (email: minoru.kubo@riken.jp)

X-ray free electron lasers (XFELs) have opened a new avenue for protein X-ray crystallography. XFELs supply ultrabright femtosecond (fs) X-ray pulses; thus a diffraction image can be obtained with a single shot from a micrometer-sized crystal. A new experimental technique using XFELs is serial fs crystallography (SFX), in which single-shot diffraction images are collected in series from a continuous flow of micro-crystals with random orientation[1, 2]. Because the fs exposure time is short enough to obtain diffraction before the onset of significant radiation damage ("diffraction before destruction")[3], early SFX studies focused on determining damage-free protein structures[4–8]. The single-shot SFX technique also offers the potential for time-resolved (TR) crystallography to investigate protein structural dynamics. Recently, pump–probe TR experiments were performed on photosystem II, carbonmonoxy myoglobin, photoactive yellow protein, and bacteriorhodopsin[9–16].

Next, structural biologists focused on irreversible systems, such as the formation of reaction intermediates during enzymatic reactions. The continuous sample delivery in SFX is advantageous because TR measurements of irreversible systems always require fresh samples for each collection of data. Mixing TR-SFX has been reported as a useful technique for supplying substrates to enzymes[17, 18]. The combination of caged-substrates with the pump–probe TR-SFX technique is another crystallographic method that may be used to examine enzymatic reactions at ambient temperature.

In the present study, we demonstrate a TR-SFX experiment in conjunction with a caged-compound as a reaction trigger, using nitric oxide reductase (nor) isolated from the fungus *Fusarium oxysporum* (P450nor). P450nor is a heme enzyme that catalyzes the reduction of nitric oxide (NO) to nitrous oxide ($N_2O$) in the nitrogen cycle ($2NO + NADH + H^+ \rightarrow N_2O + NAD^+ + H_2O$)[19]. Because $N_2O$ is the main ozone-depleting substance and a greenhouse gas[20], the NO reduction reaction mechanism that produces $N_2O$ has received increasing attention. Based on spectroscopic studies[21–23], we proposed the following reaction mechanism of P450nor (Fig. 1). In the resting state, the enzyme has a ferric heme with a water molecule and a Cys thiolate as iron axial ligands. In the first reaction, the water molecule at the sixth coordination site is displaced by NO, forming the ferric NO complex as an initial intermediate. The ferric NO complex is then reduced with hydride ($H^-$) from NADH, producing the second intermediate, intermediate-I (*I*). *I* is a two-electron reduced product of the ferric NO complex present in a singly or doubly

protonated form. Finally, *I* reacts with a second NO to generate $N_2O$. The reaction mechanism of P450nor is supported by theoretical and model-compound studies[24–27], but there are no TR structural studies.

As a reaction trigger for the TR-SFX experiment, we use caged-NO[28], which quantitatively releases NO on the microsecond time scale upon ultraviolet (UV) illumination (Fig. 2), and characterize the ferric NO complex of P450nor at ambient temperature upon caged-NO photolysis. We also characterize the ferric NO complex of P450nor using synchrotron crystallography at SPring-8 and another XFEL crystallographic technique known as serial fs-rotational crystallography (SF-ROX)[29, 30]. This technique can use large single crystals with controlled orientation. By comparing these data, we assess the properties of the ferric NO-bound structure of P450nor.

## Results

**Enzymatic reaction in the crystalline phase.** Prior to TR-SFX, we tracked the P450nor-mediated NO reduction reaction with visible and infrared IR absorption spectroscopies at 293 K to investigate the reaction kinetics in the crystalline phase. We prepared two P450nor micro-crystal systems: one containing only caged-NO (MC-1) and the other containing caged-NO and NADH (MC-2). The quantum yield of NO release from caged-NO is 1.4 with excitation at 308 nm (Supplementary Fig. 1). The yield is >1 because one caged-NO releases two NO molecules. The TR spectral changes in the MC-1 and MC-2 systems induced by UV pump illumination at 308 nm are shown in Fig. 3a, b, respectively. The spectra of the MC-1 system exhibit a positive difference band at 437 nm, which arises from the Soret band of the ferric NO complex[21], indicating that NO released from the caged-NO binds to heme. In the MC-2 system, after the appearance of a 437 nm band at 20 ms, another positive difference band was observed at 450 nm with the concomitant disappearance of the 437 nm band on the second time scale. The 450 nm band indicates the formation of *I*[21] after the reaction of the ferric NO complex with NADH. In the MC-2 reaction system, we also detected the N-N stretching band of $N_2O$ at 2228 cm$^{-1}$ after 308 nm pump illumination at 293 K (Fig. 3c), demonstrating that P450nor catalyzes the NO reduction and produces $N_2O$ in the crystalline phase. These spectroscopic results confirm that the $N_2O$ product was generated by the crystalline reaction system of P450nor through the formation of the ferric NO complex and *I*, as observed in the solution reaction. However, the reaction rate of crystal *I* formation is approximately two orders of magnitude slower than in solution[21]. The rate difference is likely caused by the slow accommodation of NADH at the active site of P450nor due to the crystal packing effect described below. Therefore, we focus on analyzing the ferric NO complex in the following TR-SFX experiment.

**Structures determined by TR-SFX at ambient temperature.** We performed TR-SFX using a lipidic cubic phase (LCP) injector[31] with hydroxyethyl cellulose matrix[32] as a novel viscous carrying medium, which achieved a slow, stable sample stream (measured

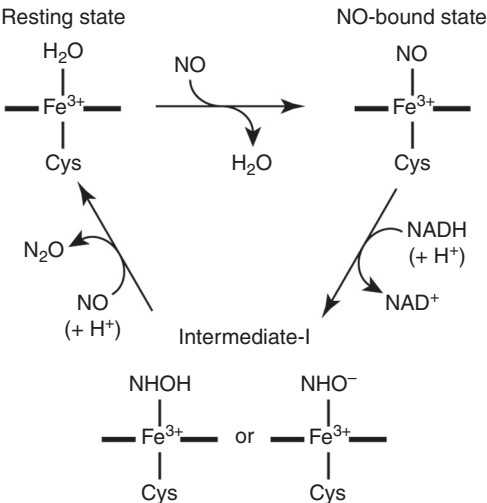

**Fig. 1** Reaction cycle of P450nor. P450nor reduces NO to $N_2O$ through the NO-bound state and intermediate-I

**Fig. 2** Caged-NO photolysis. One caged-NO releases two NO molecules upon UV light illumination

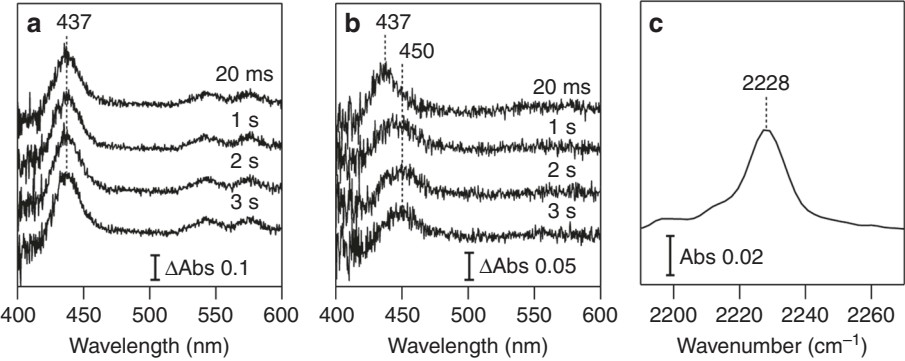

**Fig. 3** Crystal spectroscopy of ferric P450nor. TR-visible absorption difference spectra of **a** MC-1 and **b** MC-2 after caged-NO photolysis. The difference was calculated by subtracting the spectrum recorded prior to photolysis. **c** Static IR spectrum of MC-2 after caged-NO photolysis. All the measurements were performed in the presence of the SFX carrying medium (hydroxyethyl cellulose matrix) at 293 K

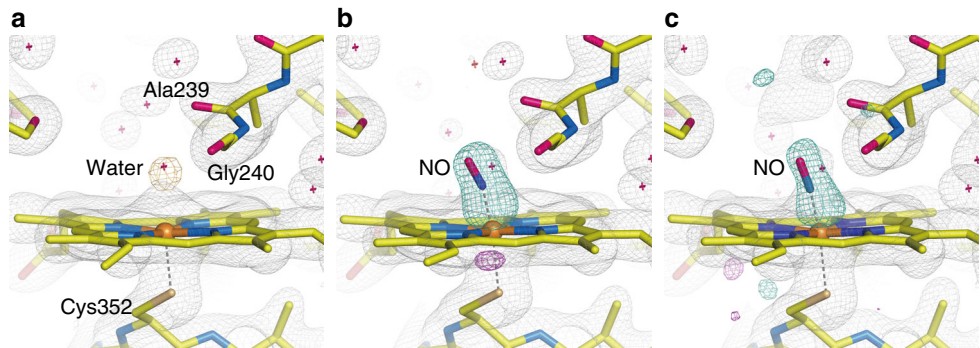

**Fig. 4** SFX structures of P450nor. **a** Resting-state structure. **b**, **c** Transient structures at 20 ms after caged-NO photolysis in the **b** absence and **c** presence of NADH. The $2F_o-F_c$ maps are shown in gray and contoured at $1.2\sigma$. The $F_o-F_c$ map is shown in orange and contoured at $4.0\sigma$ in **a**, whereas the $F_o$("Light") $-F_o$("Dark2") difference Fourier maps are shown in turquoise (positive) and magenta (negative) and contoured at $6.5\sigma$ in **b** and $3.2\sigma$ in **c**. All data were taken at ambient temperature. In **a**, the structure using the "Dark2" data of MC-2 is presented

velocity $4.6 \pm 0.3$ mm s$^{-1}$), ensuring a measurable time window up to 20 ms. However, this carrying medium is not fully transparent in the UV region and decreases the excitation efficiency of caged-NO by ~33% at 308 nm (Supplementary Fig. 1). The XFEL pulse illuminated the sample stream at 30 Hz, whereas the UV pump pulse illuminated it at 10 Hz. Thus we obtained a diffraction image sequentially from the P450nor micro-crystal upon pump illumination ("Light"), followed by two diffraction images without the pump illumination ("Dark1" and "Dark2") with a time interval of 33.3 ms. NO is only released with pump illumination; thus we can determine the structure of P450nor in the resting state from the "Dark" data. For our analysis, the "Dark2" data were used because the "Dark1" data were affected by the pump light (see Methods section). The structure of the resting enzyme consists of two molecules (A and B chains) in the asymmetric unit. The structure of the active site of the resting enzyme, as obtained by the SFX technique (SFX structure), is illustrated in Fig. 4a. In this structure, a water molecule is bound to the ferric heme as a sixth ligand, and its occupancy is 0.7 and 0.9 for the A and B chains, respectively. The coordination of the water molecule in resting P450nor is consistent with the reported structure determined by synchrotron X-rays (synchrotron structure) at cryogenic temperatures[33].

We then determined the structure of the ferric NO complex, an initial intermediate in the P450nor reaction, using the "Light" data because NO binds to the heme upon photo-irradiation of the caged-NO, as shown in the TR-spectroscopic results. Using the MC-1 system in the absence of NADH, the structure at 20 ms after caged-NO photolysis was determined at 2.1 Å resolution at ambient temperature (Fig. 4b). The $F_o$("MC-1 Light")$-F_o$("MC-1 Dark2") difference map showed a strong positive electron density at the heme distal coordination sphere, which was assigned to the bound NO. A positive density above the iron and a negative density between the iron and Cys352 indicated that the iron atom was pulled toward the distal side of the heme by NO binding. The NO molecule in this state was refined for the N-O bond length with a restraint at 1.15 Å as a target value (corresponding to neutral radical NO) because IR spectroscopy of the ferric NO complex revealed an NO stretching frequency of 1853 cm$^{-1}$, which is characteristic of neutral radical NO (Supplementary Fig. 2)[34]. In both the A and B chains, the occupancy of the bound NO is 50%, whereas a water molecule remains at 50% occupancy. Generating the ferric NO complex with full NO occupancy was difficult because the UV pump energy had to be reduced to prevent laser damage to the crystal (Supplementary Fig. 3). It is worth noting that, although we used the crystals of 20–50 μm to take advantage of the resolution of X-ray diffraction, the use of smaller crystals is an effective way to improve the NO occupancy without increasing the pump photon density. A higher caged-NO concentration may also improve the NO occupancy but could produce NO gas bubbles in the crystal due to the low solubility of NO in water. Therefore, given the required crystal size, the present experimental conditions were a practical choice for performing the TR-SFX experiment. The NO is bound to the iron in a slightly bent form with a Fe-NO distance of 1.67 Å and a Fe-N-O angle of 158°. In addition, positive and negative difference densities were observed around Ala239, indicating steric repulsion between the bound NO and the main-chain C=O of Ala239

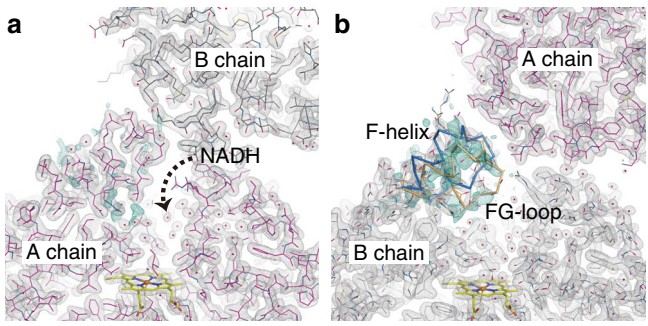

**Fig. 5** Structure of the NADH channel entrance in the **a** A and **b** B chains of resting P450nor at ambient temperature. The $2F_o–F_c$ maps are shown in gray and contoured at 1.0$\sigma$. The $F_o–F_c$ positive maps are shown in turquoise and contoured at 2.3$\sigma$. The structure factor $F_c$ was calculated from the open form (blue C$\alpha$ trace) of the NADH channel. The refined structure of the closed form is shown in orange (40% occupancy). The structures using the "Dark2" data of MC-2 are presented

(Supplementary Fig. 4). A slight movement of Ala239 by 0.2 Å was detected from the refined structure.

Using the MC-2 system in the presence of NADH, we also obtained a positive electron density at the sixth coordination site 20 ms after caged-NO photolysis in the $F_o$("MC-2 Light")–$F_o$("MC-2 Dark2") difference map (Fig. 4c), which was also assigned to the NO bound in a slightly bent form, as observed in the MC–1 system. The electron density of NO was weak compared to that of MC-1, probably because the excitation efficiency of caged-NO was lower due to the absorption of NADH in the UV region. However, no electron density of NADH was detected. There was no detectable effect of the presence of NADH on the protein conformation of P450nor or the NO coordination geometry during the 20 ms time frame.

No differences were observed among the protein moieties of all P450nor structures obtained in the TR-SFX experiments at ambient temperature. However, in all structures, the $F_o–F_c$ map indicated open and closed multiple conformations for the F-helix and FG-loop region (Fig. 5), which is the entrance of the channel for NADH to the heme pocket[35]. These types of multiple conformations in the NADH channel have never been reported[33, 36], which can be explained by the difference in space group between the used micro-crystals here and the previously used large crystals (Supplementary Fig. 5).

**NO coordination geometry determined by cryo-crystallography**. The ferric NO-bound structure obtained by TR-SFX (TR-SFX structure) using MC-1 at ambient temperature was compared with the structures obtained by SF-ROX and synchrotron crystallography. These methods can use large single crystals at cryogenic temperature and produce high-resolution structures, which provide more details of the geometry of the heme–NO moiety. The ferric P450nor crystals were soaked in an NO-saturated buffer in the absence of NADH for use in the crystallographic experiments at 100 K. Figure 6 shows the SF-ROX structure and two synchrotron structures with X-ray doses of 0.72 (low dose) and 5.7 MGy (high dose). Well-resolved electron densities for NO were observed, as shown in the $F_o–F_c$ maps. The refined NO coordination geometries obtained by TR-SFX, SF-ROX, and synchrotron crystallography are compiled in Table 1. TR-SFX and SF-ROX yield the same geometry, whereas the synchrotron data show different geometries depending on the X-ray dose. These results suggest that the NO coordination geometry does not differ substantially between ambient and cryogenic temperatures and that the TR-SFX and SF-ROX

structures are both free of radiation damage. The NO coordination in the damage-free form is characterized by a slightly bent Fe-N-O angle (158°) and a short Fe-NO bond (1.67 Å). As the X-ray dose was increased, the NO orientation became more bent (158° → 147° → 122°) and the Fe-NO (1.67 Å → 1.68 Å → 2.10 Å) and N-O (1.15 Å → 1.19 Å → 1.42 Å) bonds were elongated, as seen in the synchrotron structures. The X-ray-induced structural change was localized at the heme–NO moiety, and there was no appreciable difference in the protein structure (root-mean-square deviation of C$\alpha$ < 0.1 Å), which suggests that the photo-reduction occurs at the heme–NO moiety[37, 38]. The previously reported NO coordination structures of heme–NO species are summarized in Supplementary Table 1. Notably, a thiolate-ligated heme model complex showed a slightly bent NO coordination (160°) with a short Fe-NO bond length (1.67 Å)[39], which is consistent with the damage-free form of P450nor. A more detailed discussion is given in Supplementary Note 1.

**NO coordination geometry calculated by QM/MM**. To validate the observed NO coordination structure of ferric P450nor, we performed a geometry optimization of the damage-free structure using quantum mechanics/molecular mechanics (QM/MM). The optimization was performed for the QM region, which includes the heme–NO unit, Cys352, and the thiolate ligand loop (main chains of Ile353, Ala354, and Glu355) (Supplementary Fig. 6a). The experimental and computation structural parameters are listed in Supplementary Table 2. The calculation reproduced the experimental structural parameters, confirming that the damage-free structure is energetically suitable. The short Fe-NO bond length (1.67 Å) was reproduced with its bond order of 1.48 (Wiberg bond index), implying that the Fe-NO bond has a partial double-bond character. The slightly bent Fe-N-O geometry was also reproduced in our calculation. The potential energy surface for the Fe-N-O angle shown in Fig. 7 suggests that the Fe-N-O angle of 159° resulted in an energy minimum that was comparable to the damage-free structure.

We also performed a geometry optimization of an isolated heme model without the protein moiety (Supplementary Fig. 6b). This simplistic active site model has a relatively flat potential energy surface for a Fe-N-O angle >160° (Fig. 7). This result indicates that the thiolate trans effect[40] causes flexibility allowing the Fe-N-O bending, whereas the distal protein effect contributes to holding NO in the slightly bent conformation by producing a repulsive potential over the large Fe-N-O angle range. This potential arises from the steric repulsion between NO and Ala239 because the repulsive potential is eliminated in the QM/MM calculation where the force field derived from the Ala239 C=O group was turned off ("reduced QM/MM model" in Fig. 7). The interaction between NO and Ala239 was also identified by the non-covalent interaction analysis[41] (Supplementary Fig. 7).

## Discussion

In the present study, we demonstrated the successful application of a caged-compound for TR-SFX to characterize the ferric NO complex structure, an initial intermediate of the P450nor reaction at ambient temperature. The slightly bent NO geometry is consistent with the geometries obtained by SF-ROX and QM/MM and is damage free. The slightly bent NO coordination would be functionally relevant to the stability of the ferric NO complex in the heme pocket of P450nor that is exposed to the solvent for NADH accommodation. The ferric NO heme complexes with a linear Fe-N-O geometry have a $Fe^{2+}$-$NO^+$ electronic character and are easily decomposed by reductive nitrosylation, in which a hydroxide ion attacks the bound NO to generate nitrous acid and ferrous heme[42]. Fe-N-O bending reduces the $Fe^{2+}$-$NO^+$

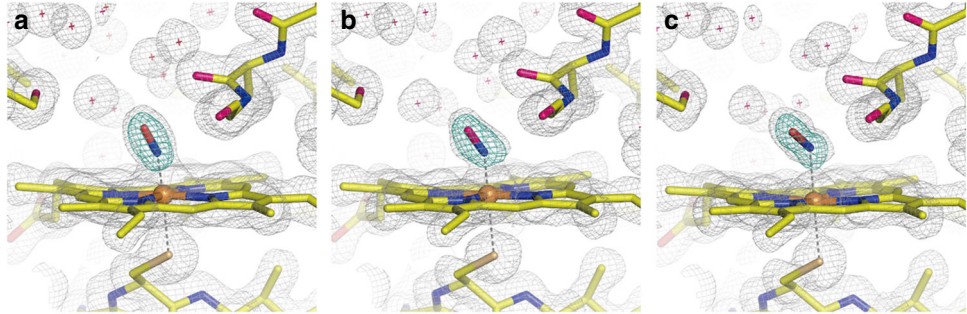

**Fig. 6** Structures of the ferric NO complex of P450nor obtained by **a** SF-ROX and **b**, **c** synchrotron X-ray crystallography. The X-ray doses in the synchrotron data collection were **b** 0.72 and **c** 5.7 MGy. The $2F_o{-}F_c$ maps are shown in gray and contoured at 1.2$\sigma$, and the $F_o{-}F_c$ positive maps are shown in turquoise and contoured at 7.0$\sigma$. All data were taken at 100 K in the absence of NADH

| Table 1 Geometrical parameters (in Å and °) for the NO coordination in the ferric NO complex of P450nor | | | |
|---|---|---|---|
| | SACLA TR-SFX (MC-1) | SACLA SF-ROX | SPring-8, low dose | SPring-8, high dose |
| Fe-NO | 1.67 | 1.67 | 1.68 | 2.10 |
| Fe-N-O | 158 | 158 | 147 | 122 |
| N-O | 1.15 | 1.15 | 1.19 | 1.42 |
| Fe-S | 2.30 | 2.33 | 2.35 | 2.32 |
| Fe-NA | 1.97 | 1.97 | 1.97 | 1.99 |
| Fe-NB | 2.05 | 2.05 | 2.05 | 2.02 |
| Fe-NC | 2.06 | 2.06 | 2.04 | 2.04 |
| Fe-ND | 2.00 | 1.99 | 2.00 | 2.03 |

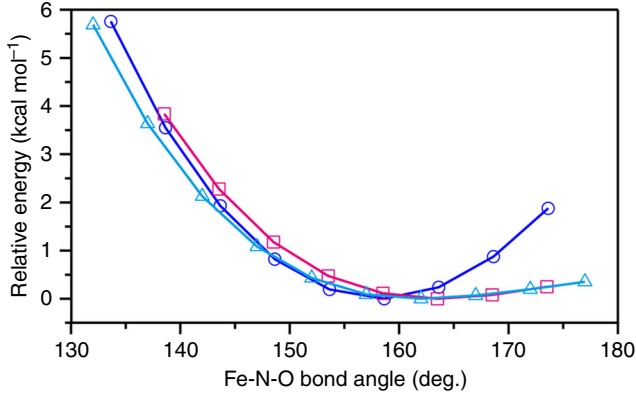

**Fig. 7** Potential energy surfaces for Fe-N-O bending. The surfaces of the QM/MM, reduced QM/MM, and isolated heme QM models are shown by blue circles, magenta squares, and cyan triangles, respectively

character; therefore, the slightly bent Fe-N-O geometry in P450nor may help to avoid reductive nitrosylation before the hydride (H⁻) attack from NADH during the NO reduction reaction.

Various caged-compounds, such as caged-ATP and photo-excitable electron donors, have been developed over the years[43, 44]. Their utility for TR crystallography was first demonstrated by a TR Laue diffraction experiment[45]. Therefore, using caged-compounds with pump–probe TR-SFX is a powerful tool for dynamic structural analyses of enzymes during their catalytic reactions in real time. However, an important finding from the present study about using this method is that the reaction in the crystalline phase must be monitored by other techniques, such as TR-spectroscopy, before the TR-SFX experiment. This is a prerequisite for determining the TR-SFX experimental conditions, as noted previously[46]. For P450nor, the rate of *I* formation in a crystal is approximately two orders of magnitude slower than that in solution. This is likely because the crystal contact decelerates NADH migration and/or NADH channel closure. Such kinetic deceleration could occur in other enzyme systems. The present pump–probe TR-SFX setup is not suitable for slow reaction phases beyond tens of milliseconds, such as *I* formation. Longer time delays between optical uncaging pulse and X-ray probe pulse are required to capture the structure of the intermediate *I*. To this end, other sample delivery methods need to be used. These could involve chip-based approaches[47, 48] or tape-based methods[49].

## Methods

**Sample preparation**. The recombinant P450nor from the fungus *Fusarium oxysporum* was expressed in *Escherichia coli* BL21(DE3) using the expression vector pRSET-C. The cell was cultured in TB medium at 30 °C. After a 4-hour culture, IPTG was added to a final concentration of 0.05 mM to induce P450nor expression, followed by overnight incubation at 30 °C. P450nor was purified from the harvested cells as follows[36]. The harvested cells were suspended with 5 mM potassium phosphate buffer (pH 8.0) containing 2.5 mg mL⁻¹ lysozyme and one tablet of

protease inhibitor cocktail (complete, EDTA-free Protease Inhibitor Cocktail Tablets, Roche Diagnostics). The cells were lysed by sonication, and the soluble fraction was obtained by centrifugation. The soluble fraction was dialyzed against 5 mM potassium phosphate buffer (pH 8.0) and was loaded onto the anion exchange column (HiPrep DEAE FF 16/60, GE Healthcare) pre-equilibrated with 5 mM potassium phosphate buffer (pH 8.0). The P450nor fraction was eluted with a linear gradient of 5–50 mM potassium phosphate (pH 6.0). After buffer of the eluted P450nor was exchanged to 10 mM potassium phosphate buffer (pH 6.0) containing 10% (v/v) glycerol by dialysis, the P450nor fraction was loaded onto the anion exchange column (HiPrep DEAE FF 16/60, GE Healthcare) pre-equilibrated with 10 mM potassium phosphate buffer (pH 6.0) containing 10% (v/v) glycerol. P450nor was eluted with a linear gradient of 10–20 mM potassium phosphate. The buffer of the purified sample with an $A_{413}/A_{280}$ ratio >1.8 was exchanged with 20 mM potassium phosphate buffer (pH 6.0) containing 10% (v/v) glycerol using a centrifugal filter unit (Amicon Ultra 30k, Merck Millipore). The sample concentration was determined by absorption spectroscopy[50].

**Crystallization**. P450nor was crystallized at 293 K using the sitting drop vapor diffusion method in drops containing ~ 25 mg mL⁻¹ sample solution (1 μL) mixed with an equal volume of reservoir solution (100 mM Bis-Tris buffer, pH 5.6–6.8, 18–34% (w/v) PEG 10,000, and 150 mM ammonium acetate). Multiple crystals were obtained within 1 day and were used for microseeding.

Plate-like single crystals for cryo-crystallography were obtained by microseeding under the same conditions as for the initial crystallization. Typically, crystals were grown to 500 × 100 × 50 μm³ in 1–2 days. To obtain the NO-bound form, the crystals were soaked in an NO-saturated reservoir solution containing 20% (v/v) glycerol for >5 min and then flash frozen with liquid N₂.

Micro-crystals for TR-SFX were obtained by batch crystallization combined with microseeding. Micro-crystals were grown in a solution prepared by mixing 5.0 mg mL⁻¹ P450nor (50 μL) and an equal volume of crystallization solution (100 mM Bis-Tris propane, pH 8.5, 34–38% (w/v) PEG 10,000, and 150 mM ammonium acetate) in PCR tubes at 293 K. Microseeds obtained by crushing the ferric P450nor crystals were also added to the crystallization solution. Micro-crystals (20–50 μm in length, < 10 μm thickness) were obtained within 1 day. Two types of micro-crystal systems were prepared for TR-SFX: one in the presence of caged-NO only (MC-1) and the

**Table 2 Statistics for TR-SFX intensity data collection and structure refinement**

| Data name | 20 ms w/o NADH "Light" | 20 ms w/o NADH "Dark2" | 20 ms w/ NADH "Light" | 20 ms w/ NADH "Dark2" |
|---|---|---|---|---|
| *Data collection* | | | | |
| Resolution (Å) | 20–2.1 | 20–2.1 | 20–2.0 | 20–2.0 |
| Highest shell (Å) | 2.12–2.10 | 2.12–2.10 | 2.02–2.00 | 2.02–2.00 |
| Space group | $P2_1$ | | | |
| Cell dimensions | $a = 54.6$ Å, $b = 102.3$ Å, $c = 73.7$ Å, $\beta = 92.6°$ | | | |
| No. of images | 308,976 | 307,266 | 127,876 | 127,897 |
| No. of hits | 147,878 | 148,786 | 48,384 | 49,115 |
| Hit rate (%) | 47.9 | 48.4 | 37.8 | 38.4 |
| No. of indexed | 108,597 | 109,519 | 34,811 | 35,309 |
| Index rate (%) | 73.4 | 73.6 | 71.9 | 71.9 |
| Measured reflections | 43,160,402 | 43,618,224 | 6,520,267 | 6,561,073 |
| Unique reflections | 47,209 | 47,209 | 54,628 | 54,628 |
| Completeness (%) | 100 | 100 | 100 | 100 |
| Redundancy | 914.2 (631.8) | 923.9 (639.2) | 119.4 (82.2) | 120.1 (56.2) |
| $CC_{1/2}$ | 0.997 (0.507) | 0.997 (0.518) | 0.981 (0.543) | 0.982 (0.591) |
| $CC^*$ | 0.999 (0.821) | 0.999 (0.826) | 0.995 (0.839) | 0.995 (0.862) |
| $R$-split (%) | 5.8 (72.7) | 5.7 (81.7) | 12.2 (78.2) | 12.0 (73.5) |
| $<I/\sigma(I)>$ | 11.3 (1.4) | 11.4 (1.4) | 6.00 (1.50) | 6.14 (1.56) |
| Wilson $B$-factor (Å$^2$) | 44.1 | 43.5 | 34.7 | 35.6 |
| *Refinement* | | | | |
| $R_{work}/R_{free}$ (%) | 14.6/19.4 | 14.6/19.3 | 14.8/19.6 | 15.1/19.7 |
| Average $B$-factors (Å$^2$) | | | | |
| Protein | 46.1 | 45.4 | 37.0 | 38.8 |
| Heme | 37.2 | 36.5 | 28.6 | 30.4 |
| NO | 40.7 | – | 33.6 | – |
| Water | 51.3 | 50.9 | 43.3 | 44.5 |
| Ramachandran (%) | | | | |
| Favored | 97.6 | 97.2 | 97.7 | 97.7 |
| Allowed | 2.2 | 2.6 | 2.1 | 2.1 |
| Disallowed | 0.1 | 0.1 | 0.1 | 0.1 |
| R.m.s. deviation from ideal | | | | |
| Bond length (Å) | 0.007 | 0.008 | 0.007 | 0.007 |
| Bond angle (°) | 0.882 | 0.964 | 0.884 | 0.887 |
| PDB code | 5Y5K | 5Y5L | 5Y5I | 5Y5J |

Values in parenthesis are those of the highest resolution shell
Index rate is the number of indexed images divided by the number of hits

other in the presence of caged-NO and NADH (MC-2). For the MC-2 system, the P450nor micro-crystals were harvested by centrifugation and resuspended in a crystallization solution containing 20 mM caged-NO [$N,N'$-bis-(carboxymethyl)-$N$, $N'$-dinitroso-$p$-phenylenediamine][28] (Dojindo) and 140 mM NADH. The slurry was incubated at ambient temperature for >10 min. The micro-crystals were concentrated by centrifugation, and the concentrated slurry of micro-crystals (10 µL) was mixed with a 32% (w/v) hydroxyethyl cellulose (Sigma-Aldrich) solution containing 13 mM caged NO and 31 mM NADH in the crystallization solution (90 µL) (final concentration: 29% hydroxyethyl cellulose (w/v), 14 mM caged-NO, and 42 mM NADH). The MC−1 system was prepared in the same way, without NADH. The preparation of the MC-1 and MC-2 systems in the carrying medium was performed under red light to avoid photolysis of caged-NO. The micro-crystals were incubated with caged-NO and NADH for at least 1 h prior to the SFX experiments because of the time it took to mix the micro-crystals with the carrying medium and put them into the SFX injector unit.

**TR-SFX at SACLA**. A 6-ns, 308-nm pulse from an optical parametric oscillator (OPO) (NT230, EKSPLA) and a <10 fs, 8.0 keV XFEL pulse were used as pump and probe pulses, respectively. A pump beam from the laser was split into two, and the beams were focused on the sample stream at an angle of 160° to each other to excite the crystal from both the front and back and thus increase the excitation efficiency. The quantum yield of NO release from caged-NO was estimated to be 1.4. The pump energy was reduced to 15–25 µJ (0.31–0.51 nJ µm$^{-2}$) from each direction to avoid laser damage to the crystal (about 60% of the damage threshold energy). The delay time ($\Delta t$) was controlled by a pulse generator (DG645, Stanford Research Systems) with a timing jitter of ~ 0.1 ms for $\Delta t > 16.7$ ms. The repetition rate of the pump laser (10 Hz) was one third of that of XFEL (30 Hz). Thus a "Light" image followed by two "Dark" images ("Dark1" and "Dark2") were obtained sequentially. To distinguish the "Light" and "Dark" images, a portion of the pump beam (5%) was picked off, and its voltage signal detected by a photodiode was used to tag the image as "Light". The pump focal size was Ø250 µm, and its beam center was aligned 45 µm upstream from the $3 \times 3$ µm$^2$ XFEL beam. The micro-crystals in the carrier medium were delivered using an LCP injector with a 75 µm nozzle at a flow

velocity of $4.6 \pm 0.3$ mm s$^{-1}$, which was estimated by high-speed camera observations (FASTCAM SA-Z, Photron). Under these experimental conditions, when the pump pulse illuminated the sample stream for "Light" data collection, the samples for "Dark1" and "Dark2" were positioned $0.11 \pm 0.01$ and $0.26 \pm 0.02$ mm upstream from the pump beam center, respectively. Because the sample for "Dark1" was partially excited by the pump beam edge (Supplementary Fig. 8), the "Dark2" data were used for the structural analysis of resting P450nor. A vacuum device was placed below the injector nozzle to stabilize the sample stream. The temperature of the syringe was maintained at 293 K. The diffraction images were collected using a multiport CCD with a sample-to-detector distance of 50 mm.

The data processing pipeline[51] based on Cheetah[52] was used for real-time feedback during the beam time, hit finding, and sorting of "Light" and "Dark" images. CrystFEL 0.6.2 was used for processing[53]. Indexing was carried out with DirAx[54]. Integrated intensities were merged by process_hkl in the CrystFEL suite. The initial phase of the structure factors was obtained by the molecular-replacement program Phenix[55] using the P450nor structure obtained with SPring-8 as a search model. The coordinates were refined by multiple rounds of manual rebuilding using Coot[56], followed by restrained refinement using Phenix. The NO molecule was refined for the N-O bond length with a restraint at 1.15 Å as a target value based on IR data. The structure refinement of resting P450nor exhibits a water molecule bound to the ferric heme with the distance of 2.4 Å. This value was used as a target value for the Fe-O distance of the partially bound water in the refinement of the NO-bound state of P450nor. The statistics for the data collection and structure refinement are summarized in Table 2. The histograms of the unit cell parameters are given in Supplementary Fig. 9.

**SF-ROX at SACLA**. The SF-ROX data were collected at BL3 of SACLA using an X-ray wavelength of 1.2404 Å (photon energy of 10.0 keV). A crystal was rotated with angle steps of 0.15 or 0.20° with a 50 µm shift between each shot to record the still diffraction images using a CCD detector (MX225-HS, Rayonix) with a sample-detector distance of 120 mm. The crystals mounted on standard magnetic pins (MiTeGen) were stored in liquid N$_2$ using Universal V1 Pucks (Crystal Positioning

**Table 3 Statistics for SF-ROX and SR intensity data collection and structure refinement**

| Data name | SACLA (SF-ROX) Damage-free | SPring-8 0.72 MGy | SPring-8 5.7 MGy |
|---|---|---|---|
| *Data collection* | | | |
| Wavelength (Å) | 1.2402 | 1.0000 | 1.0000 |
| Resolution (Å) | 25–1.50 | 30–1.50 | 30–1.36 |
| Highest shell (Å) | 1.53–1.50 | 1.53–1.50 | 1.38–1.36 |
| Space group | $P2_12_12_1$ | $P2_12_12_1$ | $P2_12_12_1$ |
| Cell dimensions | $a = 55.02$ Å, $b = 75.57$ Å, $c = 101.76$ Å | $a = 54.92$ Å, $b = 75.67$ Å, $c = 101.81$ Å | $a = 54.19$ Å, $b = 75.48$ Å, $c = 101.67$ Å |
| No. of crystals | 194 | 4 | 1 |
| No. of images | 3,087 | 160 | 360 |
| Exposure time (/flame) | < 10 fs | 4 s | 2 s |
| Measured reflections | 3,657,093 | 376,711 | 628,572 |
| Unique reflections | 65,042 | 67,790 | 87,504 |
| Completeness (%) | 94.7 (65.8) | 99.4 (98.0) | 95.2 (98.9) |
| Redundancy | 56.2 (9.2) | 5.6 (5.2) | 7.2 (7.0) |
| $CC_{1/2}$ | 0.929 (0.602) | 0.956 (0.863) | 0.967 (0.840) |
| *R*-pim (%) | | 3.3 (23.7) | 2.4 (27.4) |
| *R*-split (%) | 17.5 (40.2) | | |
| $<I/\sigma(I)>$ | 47.0 (8.6) | 17.4 (3.3) | 15.1 (2.5) |
| Wilson *B*-factor (Å$^2$) | 22.2 | 12.3 | 12.8 |
| *Refinement* | | | |
| $R_{work}/R_{free}$ (%) | 16.9/19.6 | 16.1/18.6 | 13.3/16.6 |
| Average *B*-factors (Å$^2$) | | | |
| Protein | 24.4 | 15.1 | 16.8 |
| Heme | 23.6 | 12.6 | 12.6 |
| NO | 24.6 | 15.7 | 17.8 |
| Water | 37.0 | 31.1 | 33.9 |
| Ramachandran (%) | | | |
| Favored | 98.8 | 98.1 | 99.0 |
| Allowed | 1.2 | 1.9 | 0.96 |
| Disallowed | 0.0 | 0.0 | 0.0 |
| R.m.s. deviation from ideal | | | |
| Bond length (Å) | 0.008 | 0.010 | 0.008 |
| Bond angle (°) | 0.924 | 1.094 | 1.028 |
| PDB code | 5Y5H | 5Y5F | 5Y5G |

Systems). The magnetic pin with the crystal was mounted on a goniometer using the automatic crystal exchange system SPACE. The crystal was kept at 100 K during data collection. A total of 3482 still diffraction images were collected from 194 crystals of NO-bound P450nor prepared with NO gas in the absence of NADH. Diffraction images were indexed and integrated using the cctbx.xfel package[57, 58]. The SF-ROX data were processed as follows[59]. We modified a few lines of code to suit our purpose (the modified version is available at https://github.com/keitaroyam/cctbx_fork/tree/for_sacla_sfrox). The beamstop shadow on the detector image was excluded from integration using the mask, which was defined by XDS[60]. The beam center coordinates were prerefined using geoptimizer in the CrystFEL suite[61]. The partial intensities in diffraction images were postrefined and merged by PRIME[62]. The polarization correction in PRIME was modified to match our geometric setup. Finally partial intensities obtained from 3087 images were merged.

The crystallographic refinement of NO-bound P450nor with SF-ROX data was performed with Phenix[55]. The model of P450nor refined by SPring-8 data was used as the initial model. In the structure refinement, anisotropic temperature factors were used for the atoms of heme, NO, and Cys352 (Supplementary Fig. 10). The N-O distance was restrained by a target value of 1.15 Å with the estimated standard deviation (esd) of 0.02 Å based on the IR data. In the final stage of structure refinement, the Fe-NO distance was restrained by a target value of 1.64 Å with an esd of 0.06 Å. None of the angle restraint was imposed on the NO molecule. The statistics for the data collection and structure refinement are summarized in Table 3.

**Data collection at SPring-8**. Diffraction data of NO-bound P450nor were collected at BL26B2 in SPring-8. The low-dose data were collected from four crystals with a photon flux of $6.2 \times 10^{10}$ photons s$^{-1}$ for a beam size of $74 \times 85$ μm (full-width at half-maximum (FWHM)). The first 40 images (rotation range of 40°) from each crystal were combined as a data set. The high-dose data were composed of 360 frames (rotation range of 180°) collected from one crystal. The first image was taken from a crystal that had absorbed a dose of 2.84 MGy, and the last image was taken after a dose of 5.68 MGy, with a photon flux of $2.4 \times 10^{10}$ photons s$^{-1}$ for a beam size of $67 \times 53$ μm (FWHM). The dose was calculated using RADDOSE[63]. Integration and scaling were performed with HKL-2000[64]. The coordinates were

refined by multiple rounds of manual rebuilding using Coot followed by restrained refinement using Phenix.

**TR-Visible absorption spectroscopy**. The 6-ns, 308-nm output from an OPO (NT230, EKSPLA) and the microsecond Xe-flash lamp (L11316–11–11, Hamamatsu Photonics) were used as pump and probe pulses, respectively. The delay time was controlled by a pulse generator (DG645, Stanford Research Systems) with a timing jitter of ± 20 ns. A portion of the probe beam (5%) was picked off and used to correct the pulse-to-pulse fluctuation of the probe light intensity. The pump light illuminated the sample from two directions. The pump and probe beam sizes were Ø300 and Ø40 μm, respectively, and the pump energy was 36 μJ (0.51 nJ μm$^{-2}$) from each direction. The spectra were detected by using a fiber-coupled spectrometer (Flame-S, Ocean Optics). The MC-1 or MC-2 slurry of P450nor in the presence of the hydroxyethyl cellulose medium was packed between quartz windows with a 100-μm-thick spacer and maintained at 293 K. After a crystal had been pump-illuminated, it could not be used again for data accumulation due to caged-NO consumption. Accordingly, the spectra were measured with a single shot at 1 Hz.

**IR spectroscopy**. IR spectra were measured in transmission mode using SPring-8/BL43IR equipped with an FTIR microspectrometer (Vertex 70 and Hyperion, Bruker) with a glover-lamp source. The MC-1 or MC-2 slurry of P450nor in the presence of the hydroxyethyl cellulose medium was packed between CaF$_2$ windows with a 100-μm-thick spacer. After UV pulse illumination at 308 nm (0.51 nJ μm$^{-2}$), a static spectrum was measured at 293 K. Note that the spectrum was the average derived from multiple micro-crystals distributed in the IR focal spot ($100 \times 100$ μm$^2$). For the N$_2$O band detection, multiple UV pulse illumination was necessary due to diffusion of N$_2$O after its generation. The spectrum of NO-bound P450nor was also measured at 100 K under the cryogenic N$_2$ gas, using a large single crystal loop-mounted on a goniometer. NO gas was used to prepare the NO-bound enzyme.

**QM/MM calculation**. The initial structure of the QM/MM model was constructed from the SF-ROX structure with compensating hydrogen atoms by VMD[65]. A heme, NO, Cys352, and main chains of Ile353, Ala354, and Glu355 were put in a QM region as the active site. An isolated heme QM model was also constructed as a simplistic active site model, which consisted of an Fe-porphyrin, NO, and

 

Cys352 side chain. The active site structures of both the models were optimized with B3LYP/def2svp in the spin singlet state, using the Gaussian 09 suites[66]. The ONIOM method[67] with an electronic-embedding scheme was used for the QM/MM model under the constraint that the MM region treated with an Amber ff96 force field was frozen at the initial structure. To estimate the steric repulsion between NO and Ala239, a QM/MM calculation was also performed on the condition that the MM force field arising from the main chain carbonyl group of Ala239 onto the active site was turned off (reduced QM/MM model).

**Data availability**. Coordinates and structure factors have been deposited in the Protein Data Bank under accession codes 5Y5F (SPring-8 low dose data), 5Y5G (SPring-8 high dose data), 5Y5H (SF-ROX), 5Y5I (SFX Light data with NADH), 5Y5J (SFX Dark2 data with NADH), 5Y5K (SFX Light data without NADH), and 5Y5L (SFX Dark2 data without NADH). The raw diffraction images have been deposited at CXIDB (http://cxidb.org) with CXIDB ID 63 (TR-SFX) and ID 64 (SF-ROX). All additional experimental data are available from the corresponding authors upon reasonable request.

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

## Acknowledgements

We thank the staff at SACLA and the members of Shiro laboratory at University of Hyogo for their technical support. We also acknowledge the computational support from SACLA HPC system and Mini-K supercomputer system. X-ray diffraction experiments were conducted at BL3 of SACLA with the approval of the Japan Synchrotron Radiation Research Institute (JASRI) (2015B8042, 2016A8052, 2016A8043, 2016B8068, 2017A8047), at BL26B2 of SPring-8 with the approval of RIKEN (20150042, 20160040), and at BL32XU and BL41XU of SPring-8 with the approval of JASRI (2014B1528, 2015A1122, 2015B2122, 2016A2555). IR spectroscopic experiments were conducted at BL43IR of SPring-8 with the approval of JASRI (2015B1247, 2016A1346, 2016B1146, 2017A1170). The QM/MM computations were performed at the Research Center for Computational Science, Okazaki, Japan. This work was supported by MEXT XFEL Priority Strategy Program (to H. Sugimoto, H.A., S.I.), MEXT KAKENHI GRANT 26220807 (to Y.S.), 17H05896 (to H. Sugimoto), JSPS KAKENHI GRANT 15H03841, 15H01055 (to M.K.), 15H00965 (to T.T.), JP26105012 (to Y.T.), JST-CREST JPMJCR15P3 (to O.S., H. Sugimoto), JST-PRESTO JPMJPR12L1 (to M.K.) JPMJPR14L9 (to K.H.), JST-Research Acceleration Program (to S.I.), and Pioneering Project "Dynamic Structural Biology" of RIKEN (to M.K. and M.Y.). R.Y. and H.T. were supported by the RIKEN Junior Research Associate Program.

## Author contributions

M.K., T.T., H. Sugimoto, and Y.S. designed the project; T. Nishida, N.S., K.O. and T.T. optimized crystallization conditions of P450nor and prepared the crystals; N.S., K.O., R.Y., M.S. and T.T. optimized buffer and crystal suspension conditions for TR-SFX; M.S. developed the sample delivery system for SFX; K.T., E.N., R.T., M. Yabashi, and S.I. developed the SFX system; T. Nomura, T.K., S.O. and M.K. designed the pump excitation scheme and aligned the pump beams for TR-SFX; T. Nishida, N.S., K.O., R.Y., M.S., T. Nakane, T.H., K.M., H. Sawai, H.T., E.M., A.Y., H. Sugimoto, T.T. and M.K. participated in the TR-SFX data collection; T. Nakane, O.N. and S.I. developed the data evaluation and hit finding programs for TR-SFX; K.H., G.U., H.M., M. Yamamoto, and H.A. developed the SF-ROX instrumentation; T. Nishida, T.T., H. Sugimoto, T.H. and M. K. participated in the SF-ROX data collection; T. Nomura, N.S., Y.I., O.S. and M.K. designed and performed single-crystal spectroscopy; Y.K. and Y.T. preformed computations; T. Nakane, K.Y. and H. Sugimoto analyzed the X-ray diffraction data; T. Nishida and H. Sugimoto refined the structures, calculated the electron density maps, and made these figures; M.K., T.T., H. Sugimoto, and Y.S. wrote the paper with input from all authors.
