## [Peer Review file · Nature Communications]

Reviewers' comments:

Reviewer #1 (Remarks to the Author):

In this paper the authors have used one of the new generation XFEL systems (this one in Japan) to use a pump/probe approach to capture the initial NO complex of P450nor. The paper does not provide much new insight into P450nor function that not already was achieved by conventional crystallography. That having been said, this paper warrants publication in Nature Com based on the technology. How to carry out XFEL experiments is rapidly evolving and papers like this that show how to do the proper experiments are important. The authors use a caged NO compound that releases NO after an initial flash followed by rapid XFEL data collection. This enables the first intermediate to be captured and a structure solved with zero x-ray damage. The authors also use QM/MM calculations to compare the geometry they see in the x-ray structure. This helps to confirm that the proper redox state has been observed in the crystal structure. Overall very well written and I have only a few comments.

1. The difference between "dark1" and "dark2" needs a little clarification. What is the time difference between dark1 and dark2? This is handled to some extent in the Methods but a few additional comments in the Results would help. Minor point.
2. One of the issues that comes up in these types of experiments is the release yield after the initial pump. Obviously this worked or they would not see the NO bound but is there any idea of the yield of NO release? Or was this simply a blind experiment? Try it out and see what happens.
3. On pg. 9 line 200 "The radiation damage was localized in the Fe-NO moiety." Damage may not be the best term. The problem with metal centers is usually reduction by the x-ray beam and not so much actual degradation.
4. On pg. 10 the comment is made that model complexes don't show so much damage because in-house x-ray sources were used. Partially correct. A critical part of the x-ray reduction issue is the solvent. It is generally considered that hydrated electrons do most of the reduction so if the water content is low or zero and the model complex crystals have very low solvent content, the x-ray induced reduction might be minimized. I am not 100% sure this is correct but the authors ought to check this out.
5. The sentence on pg 11 line 254 ".therefore, it is plausible.." This is a bit awkward. The iron going ferrous while the protein remains ferric doesn't make much sense. I think what they mean is that x-ray induced reduction at cryogenic temps. could capture the Fe(II)-NO geometry in form that would not be the same at room temp. In fact, there are some examples where an intermediate captured at cryogenic temps. relaxes after annealing (warm/cool).
6. The Discussion is too long. A rehash of the results is not required. The paragraph starting "In conclusion" is enough. I suggest they rework the Conclusions a bit and incorporate some of the main findings about P450nor specifically in a few sentences. The real punchline here is the technology which the Conclusion nicely handles.

In summary: accept with minor revisions.

Reviewer #2 (Remarks to the Author):

Summary:

The manuscript by Tosha et al describes XFEL and synchrotron studies on the NO complex of P450nor. Specifically, it reports the attempt of a time-resolved serial femtosecond crystallography (SFX) experiment to study the reaction $2\text{NO} + \text{NADH} + \text{H}^+ \rightarrow \text{N}_2\text{O} + \text{NAD}^+ + \text{H}_2\text{O}$ catalyzed by

P450nor which consists of three steps:

- 1) $\text{Fe}^{3+} + \text{NO} \rightarrow \text{Fe}^{3+}-\text{NO}$
- 2) $\text{Fe}^{3+}-\text{NO} + \text{NADH} \rightarrow \text{I} + \text{NAD}^+$; with I being an intermediate
- 3) $\text{I} + \text{NO} \rightarrow \text{Fe}^{3+} + \text{N}_2\text{O} + \text{H}_2\text{O}$

Tosha et al used a caged NO compound and UV flash photolysis for reaction initiation. Embedding small crystals in a viscous matrix and injecting them into the FEL beam, they collected two series of datasets. For the first, the microcrystals were soaked with caged NO, for the second with caged NO and NADH. In both cases, a "light" dataset, was collected ~20 ms after a UV flash to liberate NO, followed by two "dark" datasets. The first of the two dark datasets was reported to be contaminated by light.

The structure of the first light dataset (caged NO) shows a 50 % occupied NO and H₂O bound to the heme. The second light dataset (caged NO + NADH) seems to show less than 50% occupied NO (why?) and no bound NADH. The second dark dataset shows a 60/80 % occupied water molecule bound to the hemes of the two P450nor molecules in the asymmetric unit.

The structure of the NO complex determined using the light data set is compared to structures determined using NO-complexed crystals obtained by NO soaking before cryocooling and data collection by SF-ROX at SACLA or regular rotation crystallography at SPRING 8, respectively. In the latter case, two data sets were collected with different dose.

Finally, a QM/MM characterization of the NO complex is presented.

Assessment:

While potentially very interesting, the manuscript has several issues. 1) It is not clear what the story is: On the one hand it is presented as a time-resolved experiment to study the mechanism of P450nor. Given that step two of the reaction (see above) was attempted but failed, this is a progress report that would be better off in a more specialized journal. On the other hand, the story is presented as the first use of a caged compound for reaction initiation for data collection by SFX. This would be very interesting for the community. In that case, the part trying to analyze the second reaction step (see above, crystals MC-2) should be omitted. 2) Not enough details are given for either story line. 3) The manuscript is very difficult to read. It should be proofread by somebody proficient in English.

Detailed comments:

Major issues:

- Caged NO photolysis: The authors speculate that the occupancy of NO is only 50 % because of reduced photolysis efficiency of photolysis due to non-transparency of the hydroxyethyl cellulose medium (Page 6, line 138). First, this can and should be tested with the spectroscopy assay. In particular, if the manuscript should be published as a first use of a cage compound for TR-SFX. Knowing if a viscous medium, in particular such a useful one as hydroxyethyl cellulose can be used for pump probe TR-SFX will be of major interest and importance for the community. Second, assuming 100 % quantum yield for photolysis (which is very unlikely), and assuming that the crystals are ~ 30 um thick, one can calculate that there are about as many photons in the two photolysis beams as hemes in the crystal. The protein (and thus heme) concentration is about 16 mM in the crystal (i.e. higher than the concentration of caged NO !). Given the high extinction coefficient of 13,500 M⁻¹/cm the 1/e penetration depth is ~20 um. The pump power density is clearly not sufficient to generate enough NO for full occupancy. Another consideration is the relatively low solubility of NO in water (~ 2 mM). Information that is missing is thus: crystal size (< 50 um is not good enough), photolysis quantum yield, reasoning for the chosen photolysis laser power.

- Observation of open and closed conformations in the F-helix/FG-loop region in molecule B in the SFX data: The authors attribute this to the fact that the data were collected at ambient temperature and make a big point out of this, including in the abstract. First, a major difference between the room temperature data and all other published structures of P450nor, including the ones presented in this manuscript, is a difference in space group. How can the authors exclude

that this is not the reason for this observation? Second, should temperature be the reason for this observation, the PNAS publication of J. Fraser et al needs to be cited (doi: 10.1073/pnas.1111325108).

50 um crystals are large enough for synchrotron data collection. It would be important to establish the space group of the microseeded crystals used for the SFX experiments using conventional methods, both without and with cryocooling. It is conceivable that the spacegroup changes due to crystallization conditions, due to the addition of the cryoprotectant or due to cryocooling. Only a few medium resolution frames are needed for this, possibly using two small data ranges with a large gap in delta phi in between; then radiation damage is no concern/limitation.

Space group of SFX data: Given the high similarity of the unit cell length of the monoclinic SFX data and of the orthorhombic cryodata, it would be important to show a magnified view of the histograms of the unit cell parameters of the SFX data.

What was the reason not to analyze the SF-ROX data with CrystFEL but with cctbx.xfel instead?
- Why were the microcrystals not pre-incubated for a really long time with NADH?

Minor points (the numbers correspond to the line number):

SI unit: s instead of sec: fs instead of fsec

56 -> Recently, time-resolved pump probe experiments were performed on photosystem II, carbonmonoxy myoglobin, PYP and bacteriorhodopsin.

59: cite the recent mixing experiments: Kupitz et al, Structural Dynamics, Stagno et al, Nature 2017

77 "will be established by structural-based studies". Since the manuscript fails to deliver these, this should be rephrased to "and is the aim of future time-resolved structure-based methods"

79: time domain -> time scale

83: SF-ROX was coined in a News and Views and not in the paper cited

84: SF-ROX is not limited to cryo-cooled crystals, using very fast translation stages it can also be used for room temperature data collection

91: -> TR-visible and TR-IR spectroscopy

92 crystal form -> crystalline form

94 -> UV pump

101: Fig. 1b: It would be good to show more of the spectrum. How intense is the N-N stretching band? How much N2O has been generated? Were the crystals used for IR spectroscopy smaller than the ones used for SFX? Were they also embedded in the hydroxyethyl cellulose medium?

106 was likely -> is likely

116: what was the flow velocity of the stream? Was it measured? What was the observation of the Dark1 data set?

117 in an asymmetric -> in the asymmetric

119 , 120 water molecule was / occupancy was -> is

124 -> We determined ...

131 -> was assigned to bound NO

148 /149: This observation suggests .. No. This would have to come from kinetic data. Here it only shows that crystal packing limits binding.

150-158. Failed experiment. Not appropriate in main text.

172/173, abstract, 282 cite doi: 10.1073/pnas.1111325108

196 -> Here, we characterize

199 give Raddose reference

200 -> localized at

Section NO coordination by QM/MM. I would omit this part. I am not convinced that the steric argument is correct. Similar arguments had been used for the O2/CO complexes of myoglobin which turned out to be wrong. It would be interesting to calculate the Fe-N-O angle for an artificial in-silico Ala239Gly mutant. If the angle becomes larger, the authors strengthen their hypothesis of steric hindrance.

241 -> in-house X-ray source

270 -> in P450nor may help to reduce

272/272 -> free of radiation damage. In addition, it is noteworthy...

283 ... we established a method for... Not correct. The method was established a very long time ago. The first crystallographic application of photolysis of a caged compound was reported by Schlichting et al, Nature 1990. -> We demonstrated the successful application of ... for TR-SFX
288/289 Schlichting & Goody published a Methods in Enzymology article on time-resolved methods that might be useful to cite

306 dissolved -> dialyzed against?

Use the correct symbol for diameter, it is not a capital Phi.

Reviewer #3 (Remarks to the Author):

This is an outstanding and very detailed manuscript describing the unambiguous determinations of the three-dimensional structure of the first intermediate in NO reduction by the fungal NO reductase enzymes of direct relevance to the global N-cycle, namely the ferric-NO species that is then attacked by hydride to yield, eventually, N₂O gas. The lead authors (Shiro, Sugimoto, Kubo) are the intellectual leaders in the field, and have previously provided much of the guidance regarding the fungal and bacterial NO reduction pathways.

The molecular structures of the ferric-NO adducts of the fungal NOR species are presented, using the high-quality data obtained, with appropriately high scientific and intellectual confidence that should give a big boost to other researchers in this broad field. Although current dogma has previously assigned a completely linear Fe-N-O group to such ferric-NO species (based on limited information to date), the structural work described in this current manuscript for the authentic ferric-NO species without any photo-damage/reduction, complemented by QM-MM calculations, fully supports a slightly bent FeNO unit at ~160° for this thiolate-ligated ferric heme protein, reminiscent of that observed in a synthetic heme model. Further, the demonstration by the authors of a progressively bent FeNO unit during photo-reduction (from 158° to 147 to 122°), basically an overall transformation of a ferric-NO to a ferrous-NO derivative, is certainly the most definitive data provided in the literature that clearly shows how photo-reduction can and does change FeNO conformation in heme nitrosyl derivatives.

The results shown in Figure 1, namely the generation of the initial ferric-NO intermediate followed by generation of the second intermediate (I) and then the final product N₂O gas, are truly outstanding. Given that the reaction to generate I is slower in the crystal than in solution opens up additional avenues for future studies of this important reaction.

This reviewer appreciates the limitation of the current experimental set-up for the specialized techniques in terms of capturing the intermediate (I) for structure determination. In this case, however, the authors' inability to monitor the formation of I is not an issue; the demonstrated ability to obtain accurate data for the ferric-NO species free from radiation damage is, in itself, of very high value in this very important area of research.

Interestingly, the new finding of multiple conformations for the NADH channel is novel, and presents new opportunities to unravel the subsequent mechanisms of hydride reduction of the ferric-NO species.

Although NO is known to bind heme to elicit various biological effects, the exact nature of NO binding to ferric heme, especially that involved in the global N-cycle, has eluded scientists for decades. This current manuscript represents a landmark achievement that is fundamental to all of biological NO chemistry, and is of broad interest to a wide range of scientists in chemistry, biochemistry, biology, computational biophysics, environmental science, and agriculture. Given that structure-function studies are at the forefront of this area, it is imperative that this highly relevant work be published in Nature as soon as possible.

Minor suggestions:

- i. In the Introduction, and at the end of the 3rd paragraph, the authors may want to reproduce a summary scheme showing the formation of the ferric-NO species and intermediate (I). While not required, but this reviewer feels that it will assist the reader.
- ii. The reference 15 used in Supplementary Table S2 for ferrous HbNO can be updated with that for an R-state HbNO structure published recently in Nitric Oxide (2014), vol 39, pp 46-50 (PDB i.d. 4N8T).

Reviewer #4 (Remarks to the Author):

In this manuscript, the authors report new crystallographic data regarding P450nor and perform QM/MM calculations to ascertain the structure of the active site.

I have some remarks concerning these calculations.

First of all, the authors state that they did these QM/MM calculations with Gaussian09. Thus, I deduct that the ONIOM methodology was used, but I would prefer a clear statement in the "Methods" section. Moreover, and if ONIOM was indeed used, I hope electronic embedding between the MM and QM part was selected.

Second, and since calculations are dealing with iron(III) porphyrin species, I urge the authors to explain which spin states were calculated. Perhaps it is well-known for P450nor experts, but one should think to more general readers.

Third, I found interesting the idea to compute the Fe-NO bending to rationalize the experimental data. But, I also think it could be interesting to add Ala239 into the QM region if indeed steric repulsion is responsible for the observed geometry. Moreover, there is now methodologies in computational chemistry to ascertain the nature of weak interactions (like NCIPLOT for instance).

Fourth, I notice that the propionate side chains are included in the QM region of the QM/MM calculations. Since it adds two negative charges not properly compensated into the QM calculations, it is better either to remove them or to add also in the QM zone the cationic residues bridging with the heme.

In summary, the manuscript is interesting and as long as modifications will be included, I think it deserves publication.

Answers to the Reviewer's comments

Reviewer 1

[Comment 1] The difference between “dark1” and “dark2” needs a little clarification. What is the time difference between dark1 and dark2? This is handled to some extent in the Methods but a few additional comments in the Results would help. Minor point.

[Response 1] The XFEL pulse illuminated the sample stream at a 30 Hz repetition rate, whereas the UV pump pulse illuminated it at a rate of 10 Hz. Thus, the time difference between "Dark1" and "Dark2" was 33.3 ms. In the revised manuscript, this information is given in the Results section.

[Comment 2] One of the issues that comes up in these types of experiments is the release yield after the initial pump. Obviously this worked or they would not see the NO bound but is there any idea of the yield of NO release? Or was this simply a blind experiment? Try it out and see what happens.

[Response 2] We determined that the quantum yield of NO release from caged-NO is 1.4 with the 308 nm excitation by TR absorption spectroscopy using the solution sample of P450nor. The yield is greater than 1 because one caged-NO molecule releases two NO molecules. We also found that the hydroxyethyl cellulose medium (viscous carrying medium for SFX) reduces the excitation efficiency of caged-NO by ~33%, due to the non-transparency of the medium in the UV region. This information is given in the Results section and the spectroscopic data are summarized in Supplementary Fig. 1.

[Comment 3] On pg. 9 line 200 “The radiation damage was localized in the Fe-NO moiety.” Damage may not be the best term. The problem with metal centers is usually reduction by the x-ray beam and not so much actual degradation.

[Response 3] There is no actual degradation in the protein moiety. We rewrote this sentence in the Results section as follows: "The X-ray-induced structural change was localized at the heme-NO moiety, and there was no appreciable difference in the protein structure (RMSD of $C\alpha < 0.1 \text{ \AA}$), which suggests that the photo-reduction occurs at the heme-NO moiety."

[Comment 4] On pg. 10 the comment is made that model complexes don't show so much damage because in-house x-ray sources were used. Partially correct. A critical part of the x-ray reduction issue is the solvent. It is generally considered that hydrated electrons do most of the reduction so if the water content is low or zero and the model complex crystals have very low solvent content,

the x-ray induced reduction might be minimized. I am not 100% sure this is correct but the authors ought to check this out.

[Response 4] We agree with the reviewer's comment. We corrected this sentence as follows: "the structures of model complexes are much less damaged due to the use of an in-house X-ray source and no (or low) content of water responsible for reducing metal centers through forming hydrated electrons." We moved this to Supplementary Note 1.

[Comment 5] The sentence on pg 11 line 254 “..therefore, it is plausible..” This is a bit awkward. The iron going ferrous while the protein remains ferric doesn’t make much sense. I think what they mean is that x-ray induced reduction at cryogenic temps. could capture the Fe(II)-NO geometry in form that would not be the same at room temp. In fact, there are some examples where an intermediate captured at cryogenic temps. relaxes after annealing (warm/cool).

[Response 5] The reviewer's opinion is the same as our intended meaning. The damaged form could be some trapped form with the reduced iron. We removed the complex sentence and placed it with the following simple description. "The photo-reduction occurred at cryogenic temperature (100 K); therefore, it is plausible that the damaged form could capture the ferrous NO geometry that would be different at room temperature." We moved this to Supplementary Note 1.

[Comment 6] The Discussion is too long. A rehash of the results is not required. The paragraph starting “In conclusion” is enough. I suggest they rework the Conclusions a bit and incorporate some of the main findings about P450nor specifically in a few sentences. The real punchline here is the technology which the Conclusion nicely handles.

[Response 6] As the reviewer suggested, we removed the long discussion about P450nor from the Discussion section and left only a functional interpretation of the structural data, followed by the paragraph about the technology. This correction simplified the story in the manuscript.

Reviewer 2

Assessment: While potentially very interesting, the manuscript has several issues. (1) It is not clear what the story is: On the one hand it is presented as a time-resolved experiment to study the mechanism of P450_{nor}. Given that step two of the reaction (see above) was attempted but failed, this is a progress report that would be better off in a more specialized journal. On the other hand, the story is presented as the first use of a caged compound for reaction initiation for data collection by SFX. This would be very interesting for the community. In that case, the part trying to analyze the second reaction step (see above, crystals MC-2) should be omitted. (2) Not enough details are given for either story line. (3) The manuscript is very difficult to read. It should be proofread by somebody proficient in English.

[Response to comment (1)] The main story in this manuscript is the first use of a caged compound for reaction initiation in time-resolved SFX. The manuscript started by describing this in the Introduction and addressed it again in Conclusion.

The reviewer suggests that the part trying to analyze the second reaction step should be omitted. We might have emphasized this original purpose in the original manuscript. Thus, we deleted the related statements, and added the following sentence to the Results section: "Therefore, we focus on analyzing the ferric NO complex in the following TR-SFX experiment." in the last part of the paragraph "Enzymatic Reaction in the Crystalline Phase".

Following the paragraph on TR-SFX, we placed two paragraphs about using SF-ROX and QM/MM to assess the TR-SFX structure in the revised manuscript. Finally, we simplified the Discussion section, which only includes a functional interpretation of the obtained TR-SFX structure and the comments about the technology. We believe that these revisions clarify the story.

However, we think that we should not omit the experiment using MC-2. Using MC-2, we found that the enzymatic reaction in the crystalline form was substantially decelerated due to the crystal packing, which clearly demonstrates the importance of careful examination of how reactions proceed in the crystalline state. We think that this finding should be shared with the community. In addition, it is meaningful to include the experiment using a real reaction system (i.e., MC-2), which suits the purpose of this manuscript (the first use of a caged compound for a reaction system), and this does not complicate the story. Therefore, we left the experiment using MC-2 in the revised manuscript.

[Response to comment (2)] By responding to the reviewer's helpful comments (as below),

the manuscript has been improved, including the details about our TR-SFX experiment with a caged-compound.

[Response to comment (3)] The revised manuscript has been edited by a native English speaker. The corrected words are shown in green.

Detailed comments:

Major issues:

Caged NO photolysis: The authors speculate that the occupancy of NO is only 50 % because of reduced photolysis efficiency of photolysis due to non-transparency of the hydroxyethyl cellulose medium (Page 6, line 138). First, this can and should be tested with the spectroscopy assay. In particular, if the manuscript should be published as a first use of a cage compound for TR-SFX. Knowing if a viscous medium, in particular such a useful one as hydroxyethyl cellulose can be used for pump probe TR-SFX will be of major interest and importance for the community.

Second, assuming 100 % quantum yield for photolysis (which is very unlikely), and assuming that the crystals are ~ 30 um thick, one can calculate that there are about as many photons in the two photolysis beams as hemes in the crystal. The protein (and thus heme) concentration is about 16 mM in the crystal (i.e. higher than the concentration of caged NO !). Given the high extinction coefficient of 13,500 M⁻¹/cm the 1/e penetration depth is ~20 um. The pump power density is clearly not sufficient to generate enough NO for full occupancy. Another consideration is the relatively low solubility of NO in water (~2 mM). Information that is missing is thus: crystal size (< 50 um is not good enough), photolysis quantum yield, reasoning for the chosen photolysis laser power.

[Response 1] The reviewer mentioned caged-NO photolysis, and asked us about the properties of caged-NO and the experimental details.

i) Quantum yield of caged-NO

As in the response to comment 2 from reviewer 1, the quantum yield of NO release from caged-NO is determined to be 1.4 with the 308 nm excitation by TR absorption spectroscopy using the solution sample of P450nor. The details are summarized in Supplementary Figs. 1 (a) and (b). The yield is greater than 1 because one caged-NO releases two NO molecules. Thus, the caged-NO concentration (14 mM) is not low compared with the heme concentration (16 mM). Next, we investigated the generation yield of the NO-bound form using the MC-2 slurry in the presence and absence of the cellulose medium (Supplementary Figs. 1 (c) and (d)). The addition of the medium reduces the yield by ~33%, because of the non-transparency of the medium at 308 nm (~0.2 OD with an optical path length of 100 μm). Similarly, NADH can also cause a reduction in the

excitation efficiency of caged-NO in the MC-2 system compared with MC-1, due to the absorption at 308 nm, which could be a reason for a relatively weak NO density in the TR-SFX structure using MC-2. The above information is given in the Results section.

ii) Size of crystals used for the SFX experiments.

As the reviewer suggests, the crystal size is an important factor for TR-SFX. Smaller crystals are advantageous for excitation, whereas larger crystals may give better resolution for structural analysis. Thus, using 20-50 μm crystals was a practical compromise in this P450nor study.

iii) Excitation laser power

To excite caged-NO in the crystals with 100% efficiency, we should increase the pump laser power, as the reviewer suggested. However, we had to reduce the pump energy because the pump laser could damage the crystal. Based on the visible absorption spectroscopic measurements, we estimated the laser damage threshold to be $(34 + 34) \mu\text{J}$ with a $\text{Ø}250 \mu\text{m}$ pump diameter. Beyond this pump energy, the heme absorption peak intensity decreased. Therefore, we used a relatively low pump laser power for the diffraction experiments (about 60% of the damage threshold energy for safety).

iv) Solubility of NO

Finally, the reviewer pointed out the poor solubility of NO in aqueous solution ($\sim 2 \text{ mM}$), which is much lower than the concentration of heme in P450nor crystals (16 mM). Nevertheless, we observed about 50% NO density at the heme active site in the crystal by TR-SFX, indicating that NO was generated beyond 2 mM (at least 8 mM) from 14 mM caged-NO in the crystals. Probably, P450nor molecules bind quickly to NO generated from caged-NO, before the diffusion of the NO molecules out of the crystals. Although a higher concentration of caged-NO is expected to produce more NO-bound P450nor, it could produce NO gas bubbles in the crystals. Therefore, the present experimental conditions were a safe, practical choice for performing the TR-SFX experiments.

Because the non-transparency of the hydroxyethyl cellulose medium was not the critical reason for half NO occupancy, we corrected the sentence and added the above arguments to the Results section and Methods section.

[Comment 2] Observation of open and closed conformations in the F-helix/FG-loop region in molecule B in the SFX data: The authors attribute this to the fact that the data were collected at ambient temperature and make a big point out of this, including in the abstract. First, a major difference between the room temperature data and all other published structures of P450nor,

including the ones presented in this manuscript, is a difference in space group. How can the authors exclude that this is not the reason for this observation? Second, should temperature be the reason for this observation, the PNAS publication of J. Fraser et al needs to be cited (doi: 10.1073/pnas.1111325108).

50 μm crystals are large enough for synchrotron data collection. It would be important to establish the space group of the microseeded crystals used for the SFX experiments using conventional methods, both without and with cryocooling. It is conceivable that the space group changes due to crystallization conditions, due to the addition of the cryoprotectant or due to cryocooling. Only a few medium resolution frames are needed for this, possibly using two small data ranges with a large gap in $\Delta\phi$ in between; then radiation damage is no concern/limitation.

[Response 2] The reviewer suggested that we examine the space group of microcrystals and perform structural analysis using SPring-8 to determine why we observed the open/close conformations in the F-helix/FG-loop in the structure determined by SFX. Following the reviewer's comment, we collected the diffraction data from the microcrystals at a microfocus beamline, BL32XU, at SPring-8. The data showed that the space group of the micro-crystals (containing the cryoprotectant) was monoclinic, the same as for SFX, and the open/close conformations were detected even at 100 K, as shown in Supplementary Fig. 4. Thus, we observed multiple conformations because we used micro-crystals, not because of measuring at room temperature. In the revised version, we did not emphasize the open/close conformations as a new finding from the room temperature structure.

[Comment 3] Space group of SFX data: Given the high similarity of the unit cell length of the monoclinic SFX data and of the orthorhombic cryodata, it would be important to show a magnified view of the histograms of the unit cell parameters of the SFX data.

[Response 3] We added a magnified view of the histograms of the unit cell parameters of the SFX data (Supplementary Fig. 7). They indicate monoclinic crystals with no orthorhombic crystals.

[Comment 4] What was the reason not to analyze the SF-ROX data with CrystFEL but with cctbx.xfel instead?

[Response 4] The main reason why CrystFEL was not used for the SF-ROX data processing is because there were a limited number of images, which may not be sufficient for Monte-Carlo integration. We processed the data by two methods and compared the data quality by $CC_{1/2}$ and peak heights of anomalous difference Fourier map at the position of

the heme iron atom. The data processed using cctbx.xfel with Prime had $CC_{1/2}$ of 0.929 and the peak height of 13.6σ , whereas CrystFEL 0.6.1 (integration radius = 6, 7, 9; indexing method = mosflm) had $CC_{1/2}$ of 0.897 and the peak height of 10.5σ . Therefore, we chose cctbx.xfel.

[Comment 5] Why were the microcrystals not pre-incubated for a really long time with NADH?

[Response 5] The product information attached to NADH (Sigma-Aldrich) used in this work recommends that the NADH solution is freshly prepared and used promptly because NADH may be unstable in solution. Therefore, we did not pre-incubate the micro-crystals with NADH overnight. We also would like to mention the “practical” incubation time with NADH. Because it took time to mix the micro-crystals with the carrying medium, which contains NADH, and put the micro-crystals and NADH into the injector unit after 10 min incubation with NADH, the actual incubation time with NADH was at least 1 hour. We added this information to the Methods section.

Minor points (the numbers correspond to the line number):

[Comment]SI unit: s instead of sec: fs instead of fsec

[Comment] 56 -> Recently, time-resolved pump probe experiments were performed on photosystem II, carbonmonoxy myoglobin, PYP and bacteriorhodopsin.

[Comment] 59: cite the recent mixing experiments: Kupitz et al, Structural Dynamics, Stagno et al, Nature 2017

[Comment] 79: time domain -> time scale

[Comment] 92 crystal form -> crystalline form

[Comment] 94 -> UV pump

[Comment] 106 was likely -> is likely

[Comment] 117 in an asymmetric -> in the asymmetric

[Comment] 119 , 120 water molecule was / occupancy was -> is

[Comment]124 -> We determined ...

[Comment] 131 -> was assigned to bound NO

[Comment] 200 -> localized at

[Comment] 270 -> in P450nor may help to reduce

[Comment] Use the correct symbol for diameter, it is not a capital Phi.

[Response] We thank the reviewer for pointing out these corrections. The corrections have been made as suggested.

[Comment] 77 “will be established by structural-based studies”. Since the manuscript fails to

deliver these, this should be rephrased to “and is the aim of future time-resolved structure-based methods”

[Response] We rephrased this as "but there are no TR structure-based studies." This sentence does not say that we can provide an intermediate *I* structure and only introduces P450nor.

[Comment] 83: SF-ROX was coined in a News and Views and not in the paper cited

[Response] We have cited the News and Views article (Ref. 30).

[Comment] 84: SF-ROX is not limited to cryo-cooled crystals, using very fast translation stages it can also be used for room temperature data collection.

[Response] We rewrote this simply as "This technique can use large single crystals with controlled orientation."

[Comment] 91: -> TR-visible and TR-IR spectroscopy

[Response] The IR spectroscopic experiment was a static measurement after UV illumination of the crystal (i.e., after the enzymatic reaction). We rewrote this part simply as "we tracked the P450nor-mediated NO reduction reaction with visible and IR absorption spectroscopies..."

[Comment] 101: Fig. 1b: It would be good to show more of the spectrum. How intense is the N-N stretching band? How much N₂O has been generated? Were the crystals used for IR spectroscopy smaller than the ones used for SFX? Were they also embedded in the hydroxyethyl cellulose medium?

[Response] The IR spectra of micro-crystals presented in the original manuscript were measured in the absence of the hydroxyethyl cellulose medium. Therefore, we measured the IR spectra of micro-crystals in the presence of the medium, and replaced the spectroscopic data (new Fig. 1c for N₂O band observation and new Supplementary Fig. 2a for NO band observation) in the revised manuscript. The band frequency values were the same in the absence and presence of the medium.

The crystal size was the same as that for SFX. Only the spectrum shown in Supplementary Fig. 2b was measured using a large frozen crystal, which is described in the figure legend

in the revised manuscript. Because the IR focal size ($100 \times 100 \mu\text{m}$) was larger than the crystal size ($20\text{-}50 \mu\text{m}$), the micro-crystal spectrum was the average derived from multiple crystals distributed in the IR focal spot. Thus, the IR band intensity is difficult to analyze quantitatively.

In addition, N_2O dissociates from P450nor after generation, and diffuses out of the IR spot. We saw a gradual decrease in the N_2O band intensity. Therefore, multiple UV pulse illumination was needed to observe the N_2O band (i.e., after multiple turnover reactions). TR-IR spectroscopy (real time observation of N_2O generation) can overcome this problem, but TR-IR spectroscopy is not yet available for micro-crystals. Therefore, in this study, we present static IR spectroscopy to show that the NO reduction reaction can proceed in the crystalline state, but we cannot perform the quantitative analysis of N_2O generation.

We described these experimental conditions in the Methods section in the revised manuscript.

[Comment] 116: what was the flow velocity of the stream? Was it measured? What was the observation of the Dark1 data set?

[Response] We measured the flow velocity of the sample (hydroxyethyl cellulose medium containing P450nor micro-crystals and caged-NO), using a high-speed camera. The flow velocity was $4.6 \pm 0.3 \text{ mm/s}$ ($N = 4$). Originally, we expected the flow velocity to be 4.72 mm/s (calculated from the flow rate setting of $1.25 \mu\text{L}/\text{min}$ with a $\text{Ø}75 \mu\text{m}$ nozzle), but the measured velocity was slightly slower than the expected value. We have included the measured flow velocity value in the Methods section.

Under the above flow conditions, when the pump pulse illuminated the sample stream for “Light” data collection, the samples for “Dark1” and “Dark2” were positioned 0.11 ± 0.01 and $0.26 \pm 0.02 \text{ mm}$ upstream from the pump beam center, respectively. Because the pump beam radius was about 0.13 mm , the sample for “Dark1” was excited by the pump beam edge. Actually, the Dark1 data showd an electron density of NO, as shown in the figure below. Therefore, we did not use the Dark1 data set for analyzing resting P450nor.

Figure. The $F_o(\text{“MC-1 Dark1”}) - F_o(\text{“MC-1 Dark2”})$ difference map contoured at 4.0σ (green). The $F_o(\text{“MC-1 Light”}) - F_o(\text{“MC-1 Dark2”})$ difference map contoured at 7.0σ (yellow) is also shown for comparison.

[Comment] 148 /149: This observation suggests .. No. This would have to come from kinetic data. Here it only shows that crystal packing limits binding.

[Response] We do not have the kinetic data on the NADH binding as firm evidence. The suggestion that NADH would bind after NO binding is one possible interpretation. Therefore, we deleted this sentence from the revised manuscript.

[Comment] 150-158. Failed experiment. Not appropriate in main text.

[Response] This comment is related to the first comment from this reviewer about the manuscript story. We have deleted this paragraph, which has simplified the story.

[Comment] 172/173, abstract, 282 cite doi: 10.1073/pnas.1111325108

[Response] The multiple conformations in the NADH channel are not due to the room temperature analysis, but to the space group of the micro-crystals. So, we did not discuss the finding of multiple conformations as arising from room temperature analysis in the revised manuscript. Thus, the reference for the room temperature analysis is not necessary in the revised version.

[Comment] 196 -> Here, we characterize

[Response] We moved the photo-reduction discussion to Supplementary Note 1, and this sentence was deleted.

[Comment] 199 give Raddose reference

[Response] We have cited a Raddose reference (Ref. 61).

[Comment] Section NO coordination by QM/MM. I would omit this part. I am not convinced that the steric argument is correct. Similar arguments had been used for the O₂/CO complexes of myoglobin which turned out to be wrong. It would be interesting to calculate the Fe-N-O angle for an artificial in-silico Ala239Gly mutant. If the angle becomes larger, the authors strengthen their hypothesis of steric hindrance.

[Response] The steric interaction is between NO and the main chain C=O group of Ala239, so the Ala239Gly mutant can also have a steric effect on the Fe-N-O angle. Therefore, to elucidate the steric effect, the QM/MM calculation was performed on the condition that the force field arising from the C=O group of Ala239 was turned off. The potential curve calculated with this condition (reduced QM/MM model) agreed with that of the isolated heme model (new Fig. 5), and gave a larger Fe-N-O angle with the loss of repulsive potential in the large Fe-N-O angle range. This result demonstrates the steric effect of the Ala239 C=O group on Fe-N-O bending. We added the calculation result for the reduced QM/MM model in the Results section and in Fig. 5 in the revised manuscript.

[Comment] 241 -> in-house X-ray source

[Response] This word was corrected (Supplementary Note 1).

[Comment] 272/272 -> free of radiation damage. In addition, it is noteworthy...

[Response] We removed this paragraph in the revised manuscript because the multiple conformations are not the result of the room temperature analysis.

[Comment] 283 ... we established a method for... Not correct. The method was established a very long time ago. The first crystallographic application of photolysis of a caged compound was reported by Schlichting et al, Nature 1990. -> We demonstrated the successful application of ... for TR-SFX.

[Response] We rewrote this sentence as follows "In the present study, we demonstrated the successful application of a caged-compound for TR-SFX to characterize the ferric NO complex structure, an initial intermediate of the P450_{nor} reaction at ambient temperature." We also cited the reference (Schlichting et al., Nature 1990) as Ref. 43.

[Comment] 288/289 Schlichting & Goody published a Methods in Enzymology article on time-resolved methods that might be useful to cite.

[Response] The important points about TR crystallography are nicely summarized in this reference. We cited the reference as Ref. 46.

[Comment] 306 dissolved -> dialyzed against?

[Response] We exchanged the buffer for the purified sample using a centrifugal filter unit (Amicon Ultra 30k, Merck Millipore). We added this information to the Methods section as follows: "The buffer of the purified sample ... greater than 1.8 was exchanged with 20 mM potassium phosphate ...glycerol using a centrifugal filter unit (Amicon Ultra 30k, Merck Millipore)."

Reviewer 3

We thank the reviewer for his/her evaluation of our work. We revised the manuscript as suggested.

[Comment 1] In the Introduction, and at the end of the 3rd paragraph, the authors may want to reproduce a summary scheme showing the formation of the ferric-NO species and intermediate (I). While not required, but this reviewer feels that it will assist the reader.

[Response 1] We added a reaction scheme to the Introduction (Scheme 1).

[Comment 2] The reference 15 used in Supplementary Table S2 for ferrous HbNO can be updated with that for an R-state HbNO structure published recently in Nitric Oxide (2014), vol 39, pp 46-50 (PDB i.d. 4N8T).

[Response 2] We updated the table (new Supplementary Table 1 in the revised manuscript) and cited the reference (Ref. 18 of Supplementary references).

Reviewer 4

[Comment 1] First of all, the authors state that they did these QM/MM calculations with Gaussian09. Thus, I deduct that the ONIOM methodology was used, but I would prefer a clear statement in the "Methods" section. Moreover, and if ONIOM was indeed used, I hope electronic embedding between the MM and QM part was selected.

[Response 1] In our QM/MM calculation, the ONIOM method was used with an electronic embedding scheme. This methodology is clarified in the Methods section.

[Comment 2] Second, and since calculations are dealing with iron(III) porphyrin species, I urge the authors to explain which spin states were calculated. Perhaps it is well-known for P450nor experts, but one should think to more general readers.

[Response 2] The singlet spin state was calculated. This information is described in the Methods section.

[Comment 3] Third, I found interesting the idea to compute the Fe-NO bending to rationalize the experimental data. But, I also think it could be interesting to add Ala239 into the QM region if indeed steric repulsion is responsible for the observed geometry. Moreover, there is now methodologies in computational chemistry to ascertain the nature of weak interactions (like NCIPLOT for instance).

[Response 3] According to this comment, we extended the QM region to include Ala239 ("extended QM/MM model"), and evaluated the optimized NO coordination geometry. The geometrical parameters were obtained as follows: Fe-NO bond length 1.67 Å; Fe-N-O angle 160°; N-O bond length 1.15 Å. The NO coordination geometry was not changed by adding Ala239 to the QM region. We also evaluated the potential energy surface for Fe-N-O bending of the extended QM/MM model, as shown in the figure below. The repulsive potential in the large Fe-N-O angle range is underestimated, partially due to the lack of the dispersive interaction in the B3LYP functional. Therefore, we used the original QM/MM model in the revised manuscript.

Figure. Potential energy surfaces for Fe-N-O bending.

The reviewer also suggested the use of NCIPLOT to ascertain the interaction between NO and Ala239. We performed the non-covalent interaction (NCI) calculation between NO and its surroundings using the NCIPLOT, and the weak interaction between NO and Ala239 was visualized. In addition, the weak interaction between NO and a meso-carbon atom of the heme was also identified, which would be associated with the repulsive potential for Fe-N-O bending in the small Fe-N-O angle range. We added the NCI analysis to Supplementary Fig. 6.

[Comment 4] Fourth, I notice that the propionate side chains are included in the QM region of the QM/MM calculations. Since it adds two negative charges not properly compensated into the QM calculations, it is better either to remove them or to add also in the QM zone the cationic residues bridging with the heme.

[Response 4] As suggested, we relegated two propionate side chains to the MM region (the QM region is defined as shown in the figure below) and performed the QM/MM calculation. The optimized geometry of NO coordination was as follows: Fe-NO bond length 1.67 Å; Fe-N-O angle 156°; N-O bond length 1.15 Å. The NO coordination geometry was not changed by removing the propionate side chains from the QM region. The heme structure was also unchanged (Fe-NA, 2.02 Å; Fe-NB, 2.06 Å; Fe-NC, 2.01 Å; Fe-ND, 2.02 Å; Fe-S, 2.38 Å). This is because the two negative charges of the propionates are compensated for by the MM point charges of the cationic (Lys and Arg) residues around the propionates in the original QM/MM model.

The relegation of propionates (and the geometrical fixation of them) resulted in the ill-convergence of the geometrical optimization, and we could not achieve the optimized structure of the extended model, where the Ala239 was included in and propionates were excluded from the QM region. Therefore, we used the original QM/MM model in the revised manuscript.

Figure. The QM region with propionate relegation. The relegated propionate side chains are shown in tube representation.

REVIEWERS' COMMENTS:

Reviewer #1 (Remarks to the Author):

The authors have done an excellent job in responding to my review. The paper now is much tighter and the main take home message clear. An important piece of work that should be published.

Reviewer #2 (Remarks to the Author):

The manuscript by Tosha et al is very much improved and very nice and interesting. I fully support publication. Before doing so I would recommend a few minor changes. I indicated most of them in the pdf of the manuscript; I hope it is clear and readable.

I have two additional points:

1) The authors write that they had to restrict the optical laser power in order to avoid damage. It would be interesting for the community to specify how the damage was assessed/apparent. This should be added to the supplement.

I indicated that using smaller crystals would have solved the problem of not having enough photolysis photons. They authors might want to consider to add a comment along these lines to the supplement and why they could not use smaller crystals (resolution worse? Not enough x-ray photons?)

2) Line 232-234. As written this is an absolute statement. This is not true in general. As the authors nicely show their current setup is very good for short to medium time delays. Only very long time-delays need a different setup. This sentence needs to be rewritten.

Either system-specific:

Longer time-delays between optical uncaging pulse and x-ray probe pulse are required to capture the structure of the I intermediate. To this end other sample delivery methods need to be used. These could involve chip-based approaches (cite work from Robin Owen's and Dwayne Miller's groups and of Alke Meents group) or tape-based methods (cite 47). I think this would be best.

Alternative:

We have successfully demonstrated use of a caged compound in combination with SFX to solve the structure of a micro-second intermediate. To fully exploit the potential of this approach other sample delivery methods are needed that allow probing very long time-delays.

[Note: Illumination much further up the nozzle and adjusting the flow speed would also work.]

Reviewer #4 (Remarks to the Author):

I am convinced by the modifications made by the authors, especially regarding the steric hindrance due to Ala239. The propionate side chains QM/MM problem is a minor issue, that would request lengthy calculations to ascertain why ionic bonding is ill-described at the border between QM and MM regions. These calculations are out of the scope of this article.

Thus, I recommend the publication of this manuscript.

Answers to the comments by Reviewer 2

The manuscript by Tosha et al is very much improved and very nice and interesting. I fully support publication. Before doing so I would recommend a few minor changes. I indicated most of them in the pdf of the manuscript; I hope it is clear and readable.

We thank the reviewer's evaluation and suggestions for our work. We revised the manuscript as suggested. In addition to the English editing, there are three comments in the pdf from the reviewer.

[Comment 1 (page 6 in the pdf)] What kind of damage was observed at full laser power?

[Response 1] We checked the visible absorption spectrum of P450nor micro-crystals after single-shot UV pulse illumination. The laser pulse illumination with high energy ($>1.4 \text{ nJ}/\mu\text{m}^2$) resulted in a decrease in the heme Soret absorption. When we further increased the pulse energy above $\sim 4 \text{ nJ}/\mu\text{m}^2$, crystals were cracked. We added this information to Supplementary Fig. 3.

[Comment 2 (page 6 in the pdf)] Smaller crystals would have solved the problem...

[Response 2] We agree to the reviewer's comment that the use of smaller crystals is an important strategy to solve the problem of excitation efficiency. We added the following comment to the Results section: It is worth noting that, although we used the crystals of 20-50 μm to take advantage of the resolution of X-ray diffraction, the use of smaller crystals is an effective way to improve the NO occupancy without increasing the pump photon density.

[Comment 3 (page 12 in the pdf)] $F_{\text{obs}}(\text{Dark1}) - F_{\text{obs}}(\text{Light})$ would be interesting too.

[Response 3] We added the $F_o(\text{Light}) - F_o(\text{Dark1})$ and $F_o(\text{Dark1}) - F_o(\text{Dark2})$ maps to Supplementary Fig. 8.

In addition, the reviewer raised three more comments as below.

[Comment 4] The authors write that they had to restrict the optical laser power in order to avoid damage. It would be interesting for the community to specify how the damage was assessed/apparent. This should be added to the supplement.

[Response 4] As answered above (Response 1), we used static visible absorption

spectroscopy for the assessment of laser damage. The laser damage was obvious because the high power laser illumination ($>1.4 \text{ nJ}/\mu\text{m}^2$) decreased the heme Soret band. A higher pulse energy (e.g., $\sim 4 \text{ nJ}/\mu\text{m}^2$) induced crystal cracking. We added the damage assessment result by spectroscopy to Supplementary Fig. 3.

[Comment 5] I indicated that using smaller crystals would have solved the problem of not having enough photolysis photons. They authors might want to consider to add a comment along these lines to the supplement and why they could not use smaller crystals (resolution worse? Not enough x-ray photons?)

[Response 5] Our crystallization conditions of P450_{nor} give the crystals of mainly 20-50 μm and very few crystals of $<10 \mu\text{m}$. It is technically difficult for us to control the size of the crystals smaller than 10 μm at present. The crystal size is related not only to the NO occupancy but also to the SFX data resolution, which is also important to determine the ligand geometry. The resulting occupancy of 0.5 and resolution of 2.0-2.1 \AA from the present crystal size (20-50 μm) in this study is considered to be proper values.

However, we agree with the reviewer's point that the crystal size is an important factor for TR-SFX, especially when the quantum yield of caged compound is low. Thus, as answered above (Response 2), we added the following comment to the Results section. "It is worth noting that, although we used the crystals of 20-50 μm to take advantage of the resolution of X-ray diffraction, the use of smaller crystals is an effective way to improve the NO occupancy without increasing the pump photon density."

[Comment 6] Line 232-234. As written this is an absolute statement. This is not true in general. As the authors nicely show their current setup is very good for short to medium time delays. Only very long time-delays need a different setup. This sentence needs to be rewritten.

Either system-specific:

Longer time-delays between optical uncaging pulse and x-ray probe pulse are required to capture the structure of the I intermediate. To this end other sample delivery methods need to be used. These could involve chip-based approaches (cite work from Robin Owen's and Dwayne Miller's groups and of Alke Meents group) or tape-based methods (cite 47). I think this would be best.

Alternative:

We have successfully demonstrated use of a caged compound in combination with SFX to solve the structure of a micro-second intermediate. To fully exploit the potential of this approach other sample delivery methods are needed that allow probing very long time-delays.

[Response 6] Thank you for your suggestion. We wrote the former sentence in the revised manuscript.

[Comment 7] Note: Illumination much further up the nozzle and adjusting the flow speed would also work.

[Response 7] The present LPC injector at SACLA needs further improvement to achieve illumination of samples further up the nozzle.